# Ciliary exclusion of Polycystin-2 promotes kidney cystogenesis in an autosomal dominant polycystic kidney disease model

Rebecca V. Walker [1,3], Jennifer L. Keynton[1], Daniel T. Grimes [1,4], Vrinda Sreekumar[1], Debbie J. Williams[1], Chris Esapa[1], Dongsheng Wu [2], Martin M. Knight [2] & Dominic P. Norris [1]

The human *PKD2* locus encodes Polycystin-2 (PC2), a TRPP channel that localises to several distinct cellular compartments, including the cilium. *PKD2* mutations cause Autosomal Dominant Polycystic Kidney Disease (ADPKD) and affect many cellular pathways. Data underlining the importance of ciliary PC2 localisation in preventing PKD are limited because PC2 function is ablated throughout the cell in existing model systems. Here, we dissect the ciliary role of PC2 by analysing mice carrying a non-ciliary localising, yet channel-functional, PC2 mutation. Mutants develop embryonic renal cysts that appear indistinguishable from mice completely lacking PC2. Despite not entering the cilium in mutant cells, mutant PC2 accumulates at the ciliary base, forming a ring pattern consistent with distal appendage localisation. This suggests a two-step model of ciliary entry; PC2 first traffics to the cilium base before TOP domain dependent entry. Our results suggest that PC2 localisation to the cilium is necessary to prevent PKD.

---

[1] MRC Harwell Institute, Harwell Science Campus, Oxfordshire OX11 0RD, UK. [2] Institute of Bioengineering and School of Engineering and Materials Science, Queen Mary University of London, Mile End Road, London E1 4NS, UK. [3] Present address: School of Medicine, University of Maryland, Baltimore, MD 21201, USA. [4] Present address: Institute of Molecular Biology, Department of Biology, University of Oregon, Eugene, OR, USA. Correspondence and requests for materials should be addressed to R.V.W. (email: rebecca.walker@SOM.umaryland.edu) or to D.P.N. (email: d.norris@har.mrc.ac.uk)

Polycystin-2 (PC2) is an integral, multi-pass membrane protein with six transmembrane domains and a characteristic large extracellular loop (S1–S2 loop). It is a member of the transient receptor potential polycystic (TRPP) family of cation channels acting as a non-selective cation channel[1]. PC2 physically interacts with Polycystin-1 (PC1)[2–4] and the complex forms part of a signalling pathway involved in maintaining normal renal tubular development. The precise nature of this pathway is yet to be determined, but it likely involves the sensation of external stimuli such as mechanical shear stress or chemical signals within the lumina of the renal tubules[5–8]. It is currently argued that the primary cilium is involved in the sensing of these external cues[9].

When mutated, *PKD1* and *PKD2* cause autosomal-dominant polycystic kidney disease (ADPKD), a disorder characterised by a massive, cystic enlargement of the kidneys[10]. ADPKD is a leading cause of end-stage renal failure[11]. Like many other cystic renal diseases, ADPKD involves proteins that reside within or interact with the primary cilium[12]. The primary cilium is a microtubule-based, membrane-bounded organelle that projects from the apical cell surface. The Polycystins play an essential role in signalling from the primary cilium[8,13–15]. The cilium protrudes into the renal tubular lumen and is ideally positioned to sense extracellular stimuli and transduce information into the cell. Until recently, the cilium has been widely considered a specialised calcium signalling organelle; the traditional view being that calcium enters the cilium in response to external stimuli and the signal is transduced into the cell where it elicits a response[16–18]. Indeed, evidence from patients and animal models suggests that calcium signalling is an important factor in the pathogenesis of ADPKD[19–21]. However, recent work has argued that the primary cilium is not a calcium-responsive mechanosensor and that calcium signals are not relayed into the cell from the cilium[22], thus questioning the relationship between PC2 and the primary cilium in ADPKD.

Loss of cilia leads to kidney cyst development, while there exists a complex genetic relationship between the Polycystins and cilia. Ablation of both cilia and either PC1 or PC2 leads to a less severe cystic phenotype than loss of cilia or Polycystins alone[23]. This finding led to the proposition that a Polycystin-dependent cyst inhibitory signal is acting in opposition to cilia-dependent cyst promotion. However, this work was unable to assess the importance of ciliary localised Polycystins in the prevention of cystogenesis. While a long-established relationship exists between cilia and Polycystins, the importance of ciliary localisation remains to be determined.

Much evidence has pointed to the essential role of primary cilia in the pathogenesis of ADPKD[24–26]. The Oak Ridge Polycystic Kidney mouse was one of the earliest demonstrations of the importance of primary cilia in renal cyst development[27,28]. PC2 is almost ubiquitously expressed and performs an essential role within a number of subcellular compartments[29]. In complex with PC1, it has been shown to regulate cation currents in the apical plasma membrane[4] and together with $IP_3$ receptors in the endoplasmic reticulum (ER) is thought to regulate intracellular calcium release[30–32]. At the plasma membrane, PC2 is reported to act as a non-selective cation channel[33], and at the lateral and basolateral membranes, PC2 may function with PC1 in cell–cell adhesion[34]. Although the most prominent cellular localisation of PC2 is within the ER, it is unclear which pool of PC2 is most relevant to the protein's physiological function in preventing PKD.

We set out to question whether PC2 localisation to the cilium is essential to prevent polycystic kidney disease. We analyse *Pkd2lrm4*, a missense mutant variant that encodes a channel-functional[35] but non-cilia localising[36] form of PC2; we cannot distinguish the resulting embryonic cystic kidney phenotype from a *Pkd2* loss-of-function (null) allele. Decreased PC2 protein levels do not underlie the *Pkd2lrm4* phenotype and 50% of wild-type (Wt) protein is sufficient to prevent embryonic kidney cyst formation. Further, we describe a region at the ciliary base where mutant PC2 protein accumulates; we detect a similar but less intensely staining region in the Wt, suggesting that PC2lrm4 is limited from passing this point in the pathway to ciliary entry. These data strongly suggest that ciliary PC2 is necessary to prevent kidney cyst formation.

## Results

### *Pkd2lrm4* mutation causes cystogenesis in embryonic kidneys.
The *Pkd2lrm4* (E442G) point mutation was identified in an ENU-driven forward genetic screen looking for left–right (L-R) patterning mutants[37]. During the establishment of L-R pattern in the early embryo, PC2 interacts with PKD1L1[38] and the proteins are argued to function within the cilium in response to fluid flow[36]. We and others reported that PC2lrm4 fails to localise to cilia in the embryonic node[35,36] and that homozygous embryos develop oedema, heart defects and defective L-R pattern[37,38]. While *Pkd2lrm4* phenocopies *Pkd2* null alleles with respect to L-R patterning, analysis demonstrated that the mutant protein retains cation channel function[35]. Therefore, we set out to define the requirement for ciliary PC2 in the developing embryonic kidney.

Owing to lethality around embryonic day (E) 13.5–14.5 in *Pkd2lrm4/lrm4* embryos, it had previously been impossible to assess embryonic renal phenotypes. We therefore analysed *Pkd2lrm4/lrm4* embryos on several genetic backgrounds, finding that C57BL/6J extended embryonic survival to E15.5 (Supplementary Fig. 1), the stage at which embryonic renal cysts can first be assessed. Histological sections of E15.5 kidneys revealed cysts in *Pkd2lrm4/lrm4* kidneys (Fig. 1a, e, Supplementary Fig. 2), but not in *Pkd2lrm4/+* or *Pkd2+/+* kidneys (Fig. 1b, f, c, g). The *Pkd2lrm4/lrm4* cystic phenotype was fully penetrant and occurred bilaterally; most of the cysts were glomerular in nature (Fig. 1e), closely reflecting the phenotypes previously reported in *Pkd1−/−* and *Pkd2−/−* kidneys[39–42]. As a direct comparison, we assessed a null allele of *Pkd2* which revealed that *Pkd2−/−* embryonic kidney cysts were similar to those of *Pkd2lrm4/lrm4* mutants (Fig. 1d, h).

### Low PC2 abundance alone cannot explain *Pkd2lrm4/lrm4* cysts.
Since the level of PC2 protein is known to influence cyst development in ADPKD[43], it seemed possible that reduced PC2 protein abundance could underlie cyst development in *Pkd2lrm4/lrm4* embryos. Therefore, we next assessed PC2 protein levels in Wt and mutant kidneys. We collected protein from E14.5 *Pkd2lrm4/lrm4* kidneys; at this stage, mutants showed normal renal architecture, no cysts and in our analysis proved indistinguishable from Wt (Supplementary Fig. 2). Collecting protein at this stage therefore avoids any confounding effects of comparing cystic to non-cystic kidneys. Pools of three pairs of genetically matched kidneys provided sufficient material for analysis by western blot (WB), revealing an ~110 kDa PC2 band in all genotypes (Fig. 2a). Eight biological replicates (i.e. eight pools of three pairs of kidneys) were analysed for each genotype. Densitometric analysis revealed that non-cyst-forming *Pkd2+/−* embryonic kidneys express approximately half the protein level of Wt kidneys (average = 56.04 (± 19.68)% of Wt levels). In *Pkd2lrm4/lrm4* kidney samples, PKD2lrm4 was also reduced by half (average = 51.42 (± 15.65)% of Wt levels; Fig. 2b). The protein levels between samples of each genotype varied and it was evident that PKD2 levels in individual *Pkd2lrm4/lrm4* and *Pkd2+/−* samples overlap (Fig. 2b), even though *Pkd2lrm4/lrm4* kidneys always developed cysts, whereas *Pkd2+/−* kidneys never developed cysts. Therefore, in *Pkd2lrm4/lrm4* samples with a higher level of PC2 protein, the reduction in amount of total

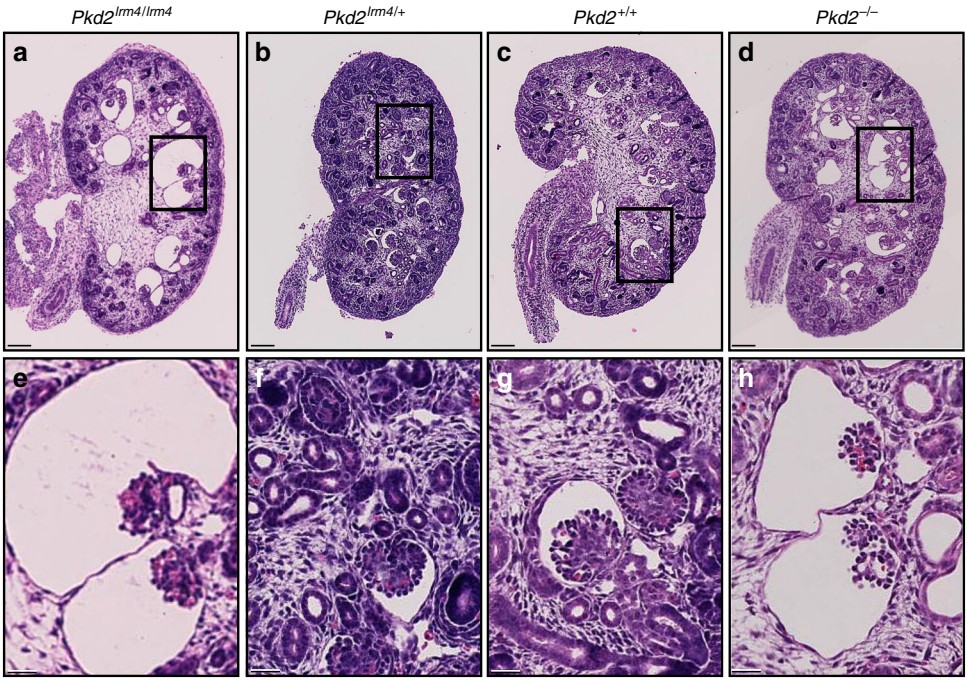

**Fig. 1** $Pkd2^{lrm4}$ causes kidney cysts. Cysts develop in $Pkd2^{lrm4/lrm4}$ kidneys (**a**) at E15.5 but not in $Pkd2^{lrm4/+}$ (**b**) or in $Pkd2^{+/+}$ (**c**). Cysts are comparable to those seen in $Pkd2^{-/-}$ kidneys at the same age (**d**). Scale bar: 100 μm. Boxed areas in **a–d** are represented in **e–h**. Glomerular cysts are visible in both $Pkd2^{lrm4/lrm4}$ (**e**) and $Pkd2^{-/-}$ (**f**). $n = 5$ E15.5 kidneys from 5 independent litters were examined for each genotype. See Supplementary Fig. 2 for representative images of embryonic kidneys. Scale bar 25 μm

PC2 protein alone cannot be responsible for the cystic phenotype. A similar reduction was evident when $Pkd2^{lrm4/lrm4}$ and Wt mouse embryonic fibroblasts (MEFs), prepared from E13.5 embryos, were compared (Supplementary Fig. 3).

To understand this reduction in PC2 levels, we examined $Pkd2$ expression by quantitative reverse transcription (qRT)-PCR. Using three sets of $Pkd2$-specific primers (Fig. 2c) we detected no significant difference between Wt and $Pkd2^{lrm4/lrm4}$ MEF samples (Fig. 2d), revealing that altered mRNA levels do not underlie the reduction in PC2 abundance. Next, we assessed whether PC2$^{lrm4}$ was being preferentially degraded by the proteasome. We incubated MEF cells with the proteasome inhibitor MG132. In Wt cells, this led to an 11% increase in PC2 levels relative to untreated cells, whereas in the $Pkd2^{lrm4/lrm4}$ MEFs a 45% increase was seen over uninhibited PC2$^{lrm4}$ levels. However, the absolute level of protein increase was similar between the two samples (Supplementary Fig. 10). Notably, proteasome inhibition was not sufficient to restore the PC2$^{lrm4}$ abundance to Wt levels, suggesting that reduced levels of PC2$^{lrm4}$ cannot simply be explained by enhanced degradation via the proteasome. We cannot rule out increased degradation of PC2$^{lrm4}$ via another pathway.

**No change in PC2$^{lrm4}$ glycosylation pattern is detected**. Glycosylation is a cell-compartment-specific, enzyme-directed modification that reflects routes taken through cellular trafficking pathways. Using in silico analysis, we were able to identify nine putative N-glycosylation sites in PC2, five falling within the first extracellular loop (S1–S2 loop; aa243–468) (Fig. 3b), in which the $Pkd2^{lrm4}$ point mutation (E442G) lies; these five sites were previously identified experimentally[44]. Since N-glycosylation and oligosaccharide processing have been implicated in protein folding and quality control, it seemed possible that the $Pkd2^{lrm4}$ mutation might result in protein misfolding, which could manifest as disrupted post-ER glycosylation.

We assessed PC2 N-glycosylation in $Pkd2^{lrm4/lrm4}$ and $Pkd2^{+/+}$ MEFs by PNGaseF digestion. PNGaseF enzymatically removes N-linked oligosaccharides from glycoproteins, cleaving between the inmost GlcNAc and asparagine residues of the oligosaccharide (Fig. 3a). Both $Pkd2^{+/+}$ and $Pkd2^{lrm4/lrm4}$ samples were sensitive to PNGaseF (Fig. 3c), demonstrating that both are N-glycosylated. As proteins traffic through the ER, they acquire specific N-glycan modifications, and these modifications are removed in a stepwise manner as the glycan travels through the Golgi. Thus Endo-H sensitivity defines the fraction of protein that has not reached the Golgi. Digestion with EndoH resulted in faster migrating bands in both samples (Fig. 3c), demonstrating that both PC2$^{Wt}$ and PC2$^{lrm4}$ protein were EndoH sensitive, consistent with them not passing the mid-Golgi.

Some studies argue that PC2 traffics to the ciliary membrane by a Golgi-independent route, leaving the ciliary portion EndoH sensitive[45]; others state that the ciliary fraction of PC2 traffics through the Golgi and becomes EndoH resistant[46]. Our results are consistent with PC2$^{Wt}$ and PC2$^{lrm4}$ being equivalently glycosylated. The EndoH sensitivity of PC2$^{lrm4}$ suggests that the major portion of PC2 protein in the mutant does not traffic through the Golgi and is likely retained in the ER, as is the case for PC2$^{Wt}$. These data suggest that there is no detectable global mis-targeting of the mutant protein, such as the aberrant cell surface trafficking, which is seen with C-terminal deletion mutant constructs[47], a result reinforced by immunofluorescent (IF) imaging of whole cells (Fig.3d, e). We saw no overall alterations of cell morphology nor indications of PC2 accumulation in the ER (Fig. 3e) of $Pkd2^{lrm4/lrm4}$ MEFs. Data exist which suggest that PC2 takes different routes to the ciliary and plasma membrane[45]. Considering that the ciliary portion is very small and may take a different route than the plasma membrane-destined PC2, it is likely that any differential glycosylation of the ciliary portion would be obscured by the larger plasma membrane portion. Therefore, we assessed ciliary localisation of PC2$^{lrm4}$.

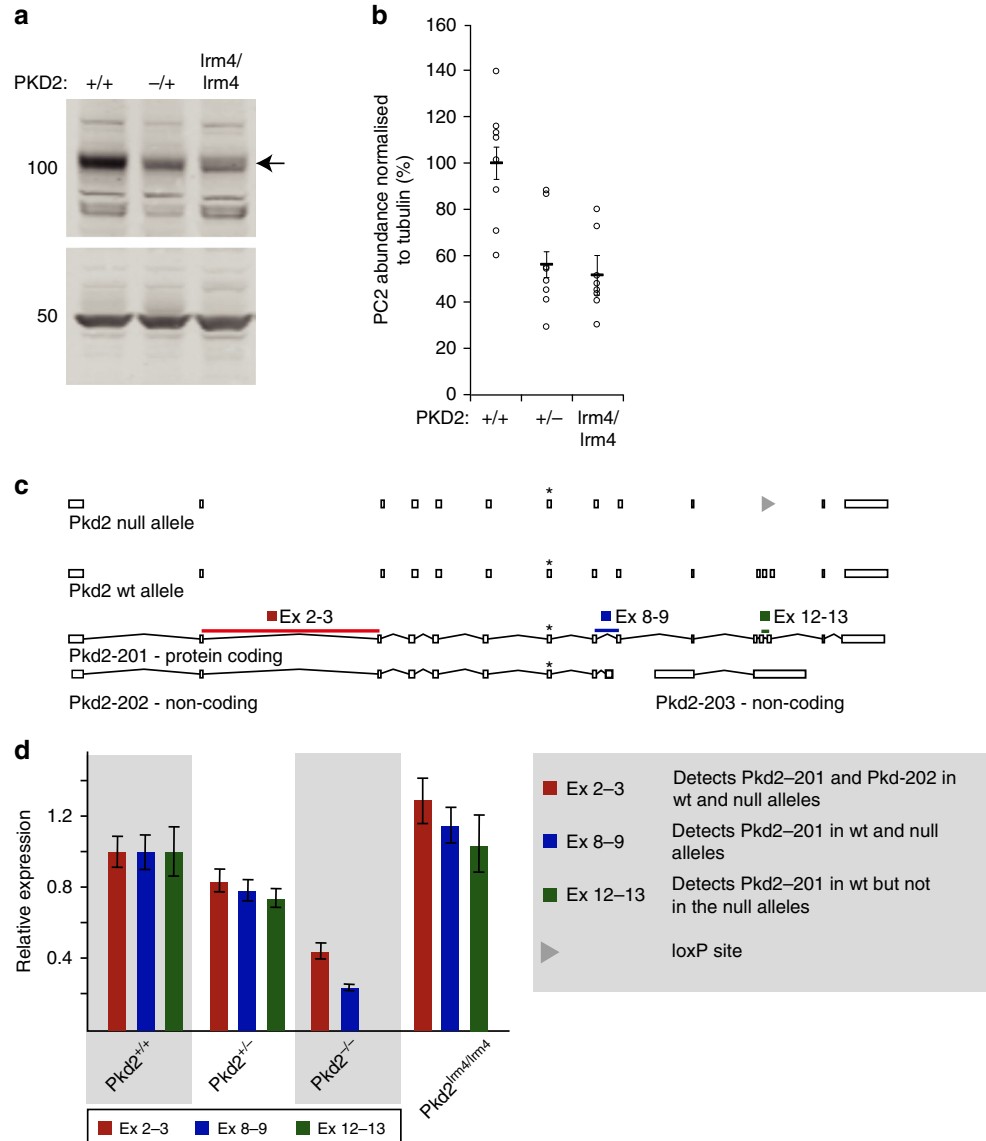

**Fig. 2** Protein and RNA expression levels of PC2. **a** Western blots showing embryonic kidney lysates from E14.5 kidneys. The arrow indicates the PC2 band at ~110 KDa (the extra bands are due to non-specific binding of antibody, see Supplementary Fig. 6). The final lane represents *Pkd2^lrm4/lrm4* kidney lysate. Tubulin loading control is shown at 50 KDa. **b** Graph represents protein levels normalised to tubulin loading control. Error bars represent SEM. $n = 8$ for each sample (pool of 3 kidney pairs). *Pkd2^+/−* (−/+) and *Pkd2^lrm4/lrm4* (lrm4/lrm4) samples are reduced by 44.0% and 48.6%, respectively, compared to Wt (+/+). **c** Schematic of PC2 gene locus showing three transcripts predicted by ensembl. The positions of primers are indicated. The position of a loxP site that replaces exons 8–9 in the null allele is marked with a triangle and the position of the lrm4 point mutation is marked with an asterisk (*). **d** qPCR of RNA extracted from MEF cells reveals that PC2 transcription is not reduced in *Pkd2^lrm4/lrm4* MEFs using primers covering three areas of *Pkd2*; Exons 2–3 (red), Exons 8–9 (blue), and Exons 12–13 (green) are the exons deleted in the null construct. *Pkd2^+/+* (+/+), *Pkd2^+/−*(+/−), *Pkd2^−/−* (−/−) and *Pkd2^lrm4/lrm4* (lrm4) MEF cells were assessed. Two sets of MEFs for each genotype were analysed and a technical repeat was performed for each sample analysed. See source data for densitometry values presented in **b**

**Cilia localisation of PC2 is disrupted in *Pkd2^lrm4***. The ciliary pool of PC2 constitutes a small proportion of cellular PC2, meaning that it may be obscured in biochemical assays. We therefore assessed the cellular localisation of PC2 by IF staining. Previous work has shown that both transgenically and endogenously expressed PC2^lrm4 failed to localise to cilia within the mouse embryonic node[35,36]. Analysis of PC2^lrm4 localisation has not been performed in other cell types. We therefore analysed PC2 localisation, using structured illumination microscopy (SIM) in primary embryonic kidney cells and MEFs. As expected, PC2^Wt protein localised along the ciliary axoneme in both cell types (Fig. 4a, d, f, h). In contrast, PC2^lrm4 protein was not detected in the axoneme (Fig. 4b, e, g, i), despite robust cell body expression. Protein was, however, seen to accumulate at the base of the cilium in both cell types (Fig. 4b, e, g, i). Co-staining with gamma tubulin, to identify the centrioles, revealed PC2^lrm4 puncta preferentially localising to one centriole, the one which forms the basal body from which the ciliary axoneme emerges (Fig. 4j, k). This staining was present in a ring of PC2^lrm4-positive puncta at the base of the cilium (Fig. 4b, e, g). An equivalent, yet often less intense, ring of PC2 was also evident in Wt cells, consistent with this region being involved in Wt PC2 trafficking

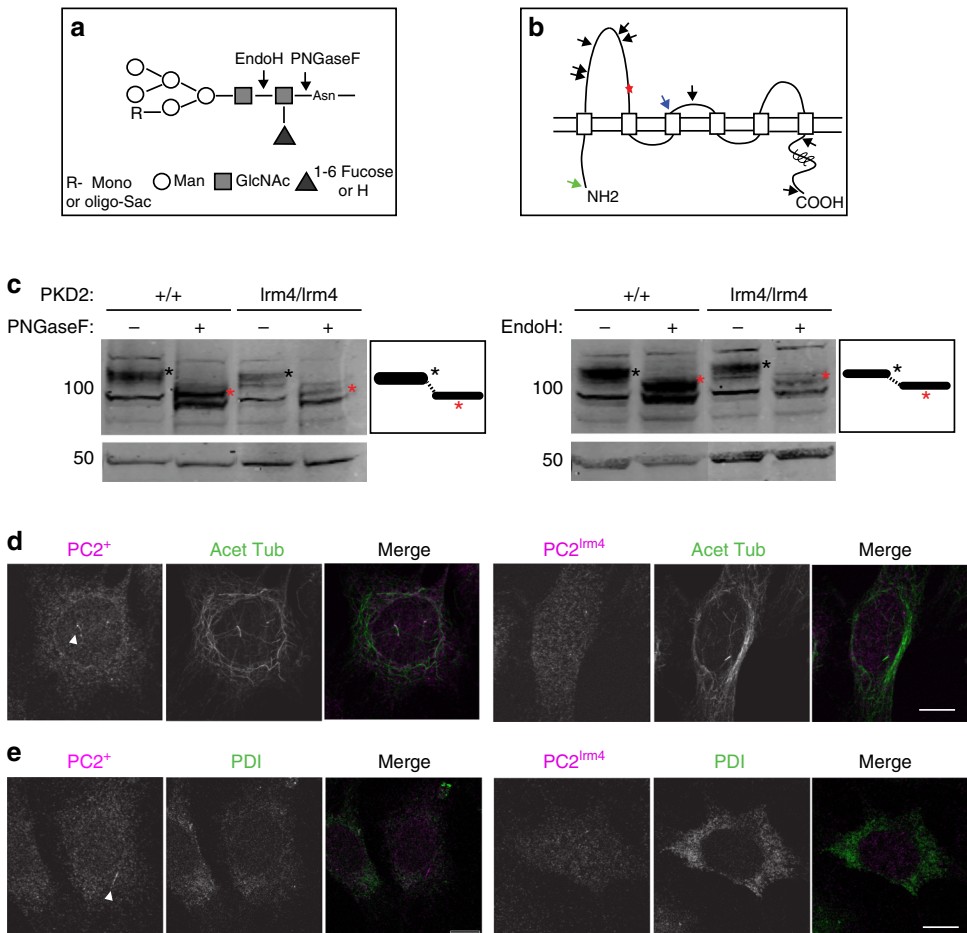

**Fig. 3** N-glycosylation pattern and cellular location of PC2. **a** Diagram depicting cleavage sites of PNGaseF and EndoH. PNGaseF cleaves between the innermost GlcNAc and asparagine of a high mannose glycoprotein. EndoH cleaves between two core GlcNAcs of an N-linked, high mannose glycoprotein. **b** Schematic of predicted glycosylation sites in PC2 (compiled from graphs in Supplementary Fig. 7). The arrows represent predicted sites of glycosylation. The black arrows represent sites shared by mouse and human. Unique sites are present in both human (green arrow) and mouse (blue arrow) sequences. The $Pkd2^{lrm4}$ mutation site is marked with a red star in the first extracellular loop. **c** MEF total cell lysates digested with PNGaseF or EndoH. To the right of each blot, a diagram represents the bands; a black asterisk (*) shows the undigested band and a red asterisk (*) indicates the enzyme-sensitive band. Both Wt (three left panels) and $Pkd2^{lrm4/lrm4}$ (right three panels) samples are PNGaseF and EndoH sensitive. Representative images of $Pkd2^{+/+}$ and $Pkd2^{lrm4/lrm4}$ cells stained with PC2(red) and **d** acetylated tubulin (green) or **e** PDI marking the ER. Cells show no overall changes in cellular PC2 localisation. PC2 is seen at the base of the cilium in mutant cells. White arrow denotes PC2 in the cilium in Wt cells. Scale bar 10 μm

(Fig. 4d, white arrows). Since PC2$^{lrm4}$ is able to reach the base of the cilium, our data imply that trafficking to the base of the cilium is unaffected but that a process involving PC2 ciliary entry is disrupted by the E442G missense mutation.

**A putative PC2-docking/sorting region at the cilium base**. The presence of a ring of PC2 protein at the ciliary base suggested that the accumulated PC2$^{lrm4}$ was likely to be associated with the mother centriole. The mother centriole is more decorated than the daughter centriole and exhibits components arranged in a radial structure (Fig. 4n). Staining for CEP164 (Fig. 4c, l), a component of the distal appendages, illuminated a region that partially overlaps with the ring of PC2 (Fig. 4m). Since the distal appendages attach to the ciliary membrane and demarcate the cilioplasm from the cytoplasm, this indicates that the accumulation of PC2 is either within trafficking vesicles or the ciliary pocket and is clearly not freely diffusing in the ciliary membrane. We have previously reported that there is no bulging at the base of these cilia[38], as can occur when proteins became trapped within the cilium[48–50]. In Wt samples, a continuous path of PC2-

positive puncta runs from the CEP164-positive distal appendages into the ciliary axoneme (Fig. 4l). The presence of PC2 in association with CEP164 is consistent with Wt PC2 residing at the distal appendages before moving into the cilium and that process being blocked for PC2$^{lrm4}$.

**Lack of ciliary PC2 in mutant cells limits PC1 ciliary entry**. Controversy exists regarding whether PC1 and PC2 enter the cilium independently or in complex with each other[8,46,47,51–54]. Since PC2$^{lrm4}$ traffics to the basal body yet is excluded from cilia, we questioned whether PC1 was excluded from the cilium. While we detect PC1 along the length of the cilium in $Pkd2^{+/+}$ cells (Fig. 5a), we could not detect PC1 in the cilia of $Pkd2^{lrm4/lrm4}$ cells (Fig. 5b), similar to PKD2$^{lrm4}$. However, we did not detect an accumulation of PC1 at the base of the cilium, as was seen for PC2$^{lrm4}$. We cannot fully rule out the possibility that weak staining was present, below the level that we were able to detect. We performed WB analysis with an N-terminal PC1 antibody to test whether PC1 abundance was reduced in $Pkd2^{lrm4/lrm4}$ cells. We found that, similar to $Pkd2^{-/-}$ samples,

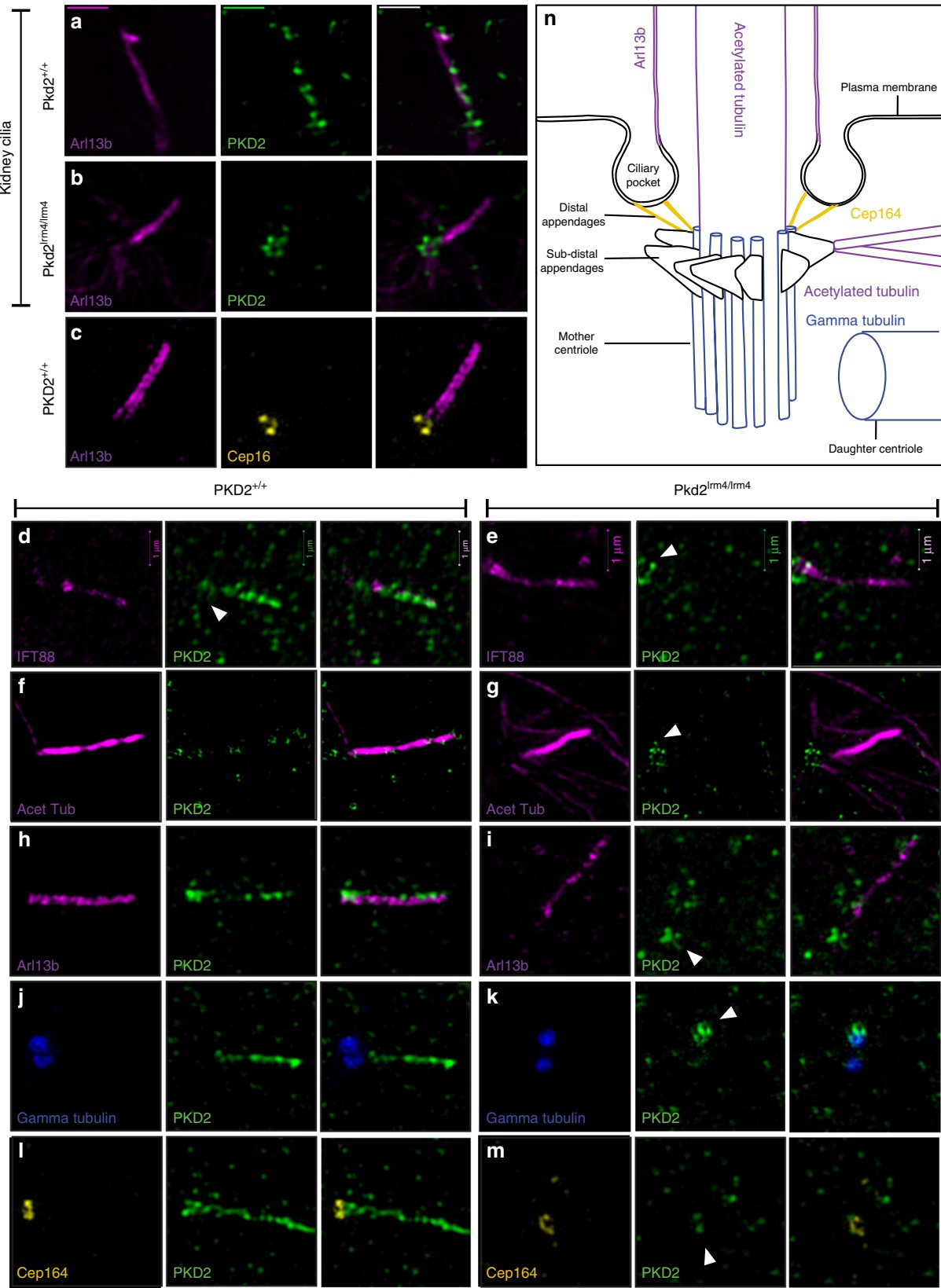

PC1 abundance is inversely proportional to PC2 abundance (Fig. 5g). Rather than being reduced in *Pkd2^{lrm4/lrm4}* cells, PC1 abundance is moderately increased, suggesting that loss of PC2 ciliary localisation impacts PC1 stability. Overall, these data indicate that, in MEF cells, PC1 ciliary localisation is dependent on PC2 ciliary entry and support the model that PC1 and PC2 enter the cilium as a complex. The question still remains whether the Polycystins traffic to the cilium in complex or meet at the base of the cilium and form a complex there before entering the cilium.

**Fig. 4** Ciliary localisation of PC2 protein. SIM images of MEF and kidney cell cilia showing PC2 localisation. **a**, **b** are representative of primary kidney cell cilia. In Wt MEF cells, CEP164, a component of the distal appendages of the mother centriole localises at the base of the cilium (**c**). In Wt kidney cilia (**a**) and Wt MEF cilia (**d**, **f**), PC2 localises along the length of the cilium. PC2 is seen in a ring structure at the base of the cilium in both Pkd2[lrm4/lrm4] kidney cilia (**b**) and MEF cilia (**e**, **g**) (white arrow). **h** Arl13b marks the ciliary membrane and PC2[+] localises along the length of the cilium and is observed more densely at the cilium base. **i** PC2 is seen as a cluster at the base of the cilium. **j** Gamma tubulin localises at the base of the cilium and PC2[+] can be seen to localise along the length of the cilium. **k** In mutant cells, a pool of PC2[lrm4] associates preferentially with the mother centriole (white arrow). In Pkd2[lrm4/lrm4] cells, PC2 clusters at the base of the cilium (white arrow). **l** CEP164 marks the distal appendages at the base of the cilium and PC2 localises along the length of the cilium as well as preferentially to the mother centriole. **m** The PC2[lrm4] cluster associates with the distal appendages. **n** The diagram indicates the position of each antibody with respect to the cilium with corresponding colours. Constituents of the ciliary axoneme or membrane: Ift88, acetylated tubulin and Arl13b are coloured magenta; PC2 is coloured green; Gamma tubulin, marking the basal body, is coloured blue; and cep164, marking the distal appendages, is coloured yellow

## Discussion

The link between cilia and kidney cysts remains enigmatic[55]. PC2 localises to cilia, as well as to the plasma membrane, the ER and intercellular junctions[56]; in theory, any or all of these locations could be important in cyst prevention. The nature of the majority of patient *PKD2* mutations and *Pkd2* models (mostly consisting of nonsense mutations or mutations resulting in complete protein loss) means that we cannot distinguish between the potentially distinct roles of PC2 in different compartments. The PC2-W414G ADPKD variant fails to localise to cilia, supporting the argument for ciliary function[57]. The mouse PC2[lrm4] protein maintains channel function but fails to localise to node cilia[35,36]. In this study, we demonstrate that PC2[lrm4] similarly fails to localise to the ciliary axoneme in embryonic kidney outgrowths and in fibroblasts. We further show that *Pkd2[lrm4/lrm4]* embryos develop kidney cysts that proved indistinguishable from those of *Pkd2[−/−]* embryos. Although we found reduced PC2 protein levels in *Pkd2[lrm4/lrm4]* mutants, this alone could not account for the cystic phenotype. Our data suggest that PC2 function is essential within the ciliary axoneme to prevent cyst development.

The nature of the PC2[lrm4] protein remains an unknown variable in this study. PC2[lrm4] retains cation channel function in a cell-free system and does not localise to cilia. However, our understanding of the TOP domain in which the mutation lies is nascent. PC2 makes both channel-forming homotetramers, as well as heterotetramic channels with PC1, yet we do not know the effect that the mutation has on the formation of these. As such, we cannot fully rule out a more complex explanation of the phenotype. It is formally possible, for example, that PC2[lrm4] protein exhibits a reduced, hypomorphic function in addition to reduced protein levels. In such a situation, the level of PC2 function might drop beneath that required to prevent cyst formation. To investigate this would require the development of a measurable assay of PC2's anti-cystic function or a correlate of this function; this in turn would require a deeper understanding of the mechanism by which PC2 prevents cyst formation.

As reduced protein levels seemed unlikely to underlie the *Pkd2[lrm4/lrm4]* phenotype, it appeared possible that disrupted cellular localisation of PC2[lrm4] results in cyst formation. The route of PC2 trafficking to the cilium is controversial, with contradictory claims about whether PC2 passes through the Golgi, as assessed by EndoH sensitivity[45,46]. We found no discernible difference between the EndoH sensitivity of Wt and *lrm4* mutant PC2, but the previously highlighted caveat of the relatively small size of the ciliary fraction of PC2 makes it uncertain whether a potential difference would be detected. The lowered expression level in *Pkd2[lrm4/lrm4]* samples reveals an altered abundance or ratio of bands compared to the Wt banding pattern. This may indicate that PC2[lrm4] has altered cellular trafficking. Since we do not see an ER accumulation of PC2, as has been reported in non-trafficking PC2 mutants, it seems most likely that PC2[lrm4] traffics similarly to Wt PC2. Indeed, we demonstrate that a ring of

PC2[lrm4] accumulates at the base of cilia. Since we observe a similar but weaker accumulation of PC2 at the base of Wt cilia, it seems most likely that this is part of a normal ciliary transport route for PC2. SIM imaging resolves this ring to comprise a series of puncta, positioned at the base of the mother centriole, associating with the CEP164-positive distal appendages. The mutant PC2[lrm4] is unable to pass beyond this region, which seems most likely to be part of a docking region involved in ciliary entry.

Recent structural characterisation of PC2 reveals that it is able to make a characteristic tetrameric TRP channel. The S1–S2 loop of PC2, however, encodes an additional "TOP domain", which sits above the channel pore[58,59]. ADPKD missense mutations cluster in the TOP domain[60], underlining its importance in PC2 function. Two highly conserved polycystin domains, PC-A and PC-B, map to this region[61,62]. While their function remains unknown, both PC2[lrm4] and W414G lie in PC-B and both affect ciliary localisation[35,36,57]. In combination with our findings, this suggests that the S1–S2 loop region is required for normal transport of PC2 into the cilium. It is interesting to speculate that this might be through interference with PC1–PC2 heterotetramer formation and that mutations in the TOP domain directly influence these interactions. Investigation of the TOP domain structure reveals that the TOP domain assumes a different conformation in membranes with different lipid compositions[59]. It is possible therefore that certain TOP domain mutations may alter the conformation sufficiently to prevent the channel from forming; such a fundamental block to complex formation could conceivably prevent PC1–PC2 complex trafficking. Conversely, mutations in the TOP domain may affect function of the channel complex, resulting in subtle alterations in channel activity that are not evident in the cell-free system in which PC2[lrm4] was previously tested. These two hypotheses could be tested if an assay existed that could distinguish homomeric, heteromeric and single subunit forms of the protein. Disrupted channel formation or function may lead to PC2 protein being sent for degradation, which would be consistent with the lower level of PC2 protein that we see in the mutant cells. The Polycystins have previously been reported to enter the cilium as a complex[54], joining together before entering the cilium[46,47,52,53]. This is indeed consistent with what we observe. However, since PC1 and PC2 traffic differently through the Golgi[45,63] and have discrete ciliary targeting signals[47,51], it is possible that the two meet at the base of the cilium before entering together. However, PC2 has been shown to localise to the cilium in cells collected from kidneys after PC1 inactivation[8] and has been noted to traffic independently of PC1 in some systems[51], suggesting that PC1 is not always essential for PC2 ciliary localisation. In this case, perhaps a homomeric interaction between PC2 molecules is sufficient to allow PC2 ciliary trafficking.

Recently, a cryo-electron microscopic structure has been solved for PC1 and PC2 in complex[64]. This reveals the two proteins in a 1:3 (PC1:PC2) ratio, forming a structure reminiscent of the PC2

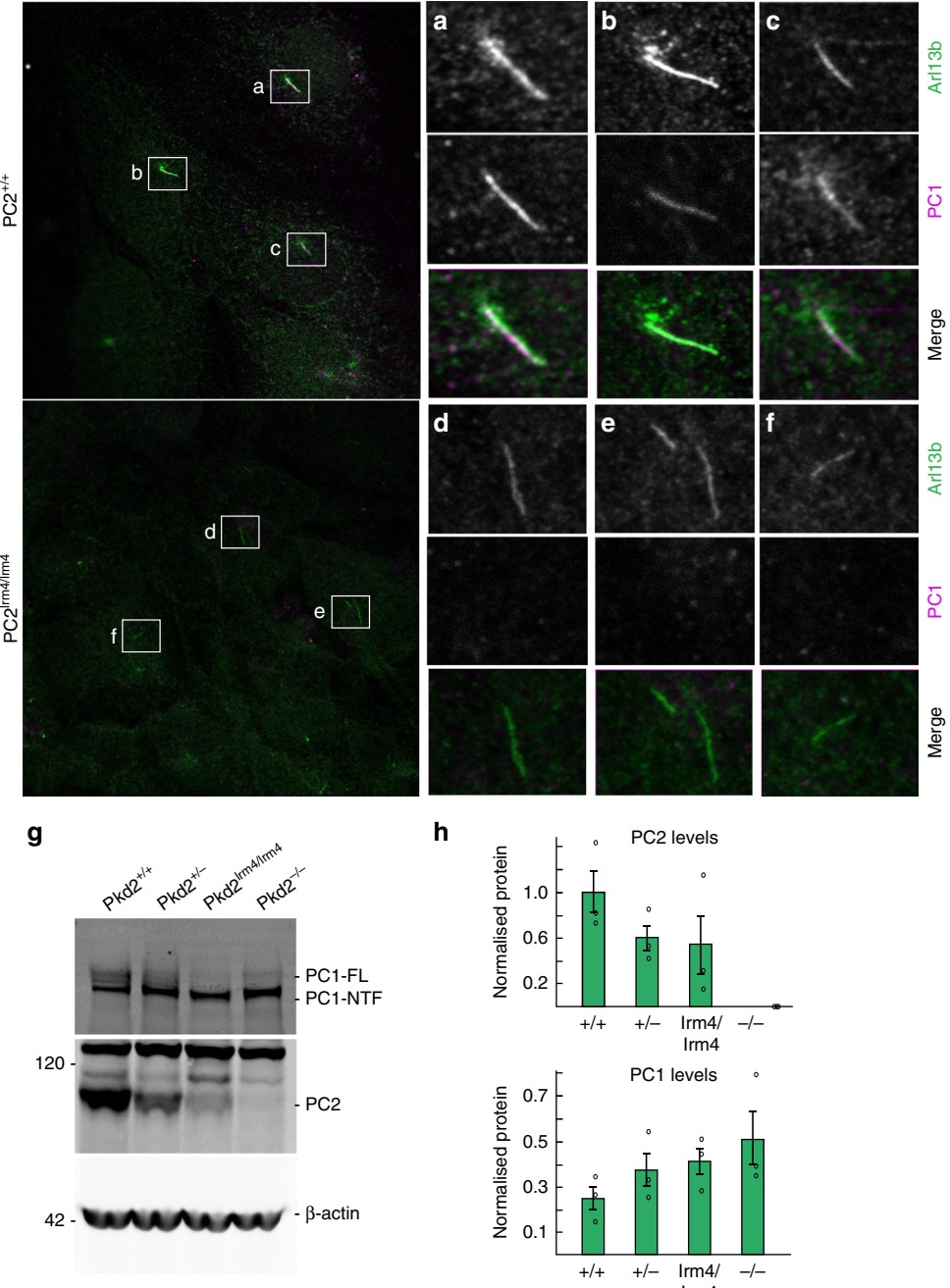

**Fig. 5** PKD1 is not detected in the cilium in *Pkd2^lrm4/lrm4* cells. In Wt MEFs, PKD1 localises along the length of the cilium. **a–f** enlarged from boxes in Merge on left. PC1 localises along the length of Wt cilia (**a–c**). In *Pkd2^lrm4/lrm4* mutant cells, PC1 ciliary localisation is no longer observed. **g** Western blot analysis of MEF samples probed with an N-terminal-directed PC1 antibody and an antibody directed against PC2. β-Actin serves as a loading control. **h** Densitometry of PC1 and PC2 bands normalised to β-actin is represented as graphs. Error bars represent SEM. See source data for densitometry values presented in **g**

tetramer; importantly, PC1 and PC2 are in contact throughout much of their length. In light of data that they can traffic independently to the cilium, this raises questions about how and where the heteromeric complex forms. Does a PC2 tetramer have one subunit replaced by PC1 at the cilium or does a stabilised PC2 trimer traffic to the ciliary base where it associates with PC1? The ring of PC2 that we observe at the base of the cilium may represent a PC1–PC2 rendezvous point, where heteromeric complexes are formed, or simply a docking site for PC2 to be loaded into the ciliary transport system. This region appears to surround the basal body around the level of the CEP164-positive distal appendages. These constitute part of the size-selective gate

at the base of the cilium, preventing large complexes from diffusing into the cilium, as well as delineating a region for vesicle docking and unloading[65]. Rab-8-positive vesicles have been reported to dock at the base of the cilium[66] and are proposed to release PC2 at the sub-distal appendages. A recent study into how proteins exit the cilium described a two-step process in which activated G-protein coupled receptors leaving the cilium first cross the transition zone into an intermediate compartment before exiting a stringent gate at the distal appendages[67]. Another study revealed that distal appendages are an essential component of the ciliary gate and identified structural components of the ciliary gate that appear to be necessary for either ciliary entry or

retention[68]. This work supports the hypothesis of a docking site for membrane proteins at the base of the cilium and is consistent with the distal appendage location of PC2$^{lrm4}$.

The mechanistic relationship between the Polycystins, cilia and cyst formation has been difficult to elucidate. Loss of Polycystins and loss of cilia are both cystogenic, suggesting that Polycystins provide a cilia-dependent anti-cystogenic signal[9]. However, loss of both cilia and Polycystins results in a partial rescue of the Polycystin cyst phenotype, suggesting that cilia lacking Polycystins drive rapid cyst growth[23]. This has led to the argument that ciliary Polycystins normally act to repress a pro-cystic ciliary signal[23]. However, the kidneys examined in these studies lacked all Polycystin function, meaning that other more complex interpretations cannot be fully excluded. The Polycystins exhibit pleiotropy within the cell: they have ciliary and cell body functions, and these functions may involve distinct pathways. Our analysis describes a ciliary excluded, yet channel functional form of PC2[35], meaning that the cell body PC2 likely still functions. We therefore provide strong evidence that cilia-localised PC2 functions as an anti-cystogenic signal. As we cannot express PC2 solely in the cilium however, we cannot exclude possible additional anti-cystogenic roles for PC2 populations that localise to different cellular compartments.

This study presents evidence that altered ciliary localisation of PC2 is sufficient to cause cystogenesis in the presence of channel-functional, non-cilia-localising protein. We show that disruption of the ciliary localising population of PC2 is sufficient to cause cyst development and that the remaining cellular portion is unable to prevent cystogenesis; we cannot, however, rule out additional anti-cystogenic activity elsewhere in the cell. Our work uncovers a putative docking/sorting region for PC2 at the base of the cilium and we suggest that transiting this region is essential for ciliary localisation and therefore is required to enable the ciliary-dependent cyst-preventing function. Moreover, both the lrm4 and W414G mutations show that the highly conserved S1–S2 loop is required for normal ciliary trafficking. The simplest explanation of this data is that integration of PC2 into the ciliary membrane is necessary to prevent kidney cyst formation.

## Methods

**Ethics statement**. All experiments were performed under the guidelines and approval of the MRC Harwell Ethics Committee and the UK Home Office; euthanasia was by cervical dislocation of adults and decapitation of embryos.

**Mice**. $Pkd2^{lrm4}$ mice were congenic on C57BL/6J. The $Pkd2^{-}$ allele, $Pkd2^{tm1.2Tjwt}$, was derived from $Pkd2^{tm1.1Tjwt}$ by Cre deletion and maintained congenic on C57BL/6J. Both male and female embryos were analysed in this study. As embryonically no difference in the severity of kidney cyst development is expected from the published data, embryos' sex was not determined.

**Cell lines**. MEFs and kidney outgrowths were produced from primary mouse tissue. Embryonic carcasses or kidneys were minced and digested in 0.25% trypsin. The digested tissue was then plated in Dulbecco's Modified Eagle's medium supplemented with foetal bovine serum. The SV40 large-T antigen plasmid was electroporated into primary MEFs to produce stable cell lines.

**Immunofluorescence**. Cultured cells were fixed with ice-cold methanol (5 min), permeabilized by 1% NP-40 (3 min), blocked in 4% bovine serum albumin (30 min) and incubated with primary antibodies in blocking buffer for 1 h at room temperature. After phosphate-buffered saline washes, cells were incubated with fluorescence-conjugated secondary antibodies for 30 min at room temperature. Cells were washed and mounted with Hydromount (National Diagnostics).

**Antibodies**. Mouse anti-acetylated tubulin (working concentration 2 μg/ml, Sigma-Aldrich, T7451); mouse anti-ARL13B (working concentration 2 μg/ml, Abcam, ab136648); mouse anti-CEP164 (working concentration 5 μg/ml, Sigma, SAB2702133); goat-anti-IFT88 (working concentration 1 μg/ml, Abcam, ab42497); mouse-anti-gamma tubulin (working concentration 1 μg/ml, Sigma, T6557); mouse anti-PC1 (working concentration 1 μg/ml for IF, 0.4 μg/ml for WB, 7E12, Santa Cruz, SC130554); and rabbit anti-PC2 (working concentration 1 μg/ml for both IF

and WB, H-280, Santa Cruz, SC25749); mouse anti-β-actin (working concentration 0.1 μg/ml for WB, Sigma, A5316); mouse anti-Tubulin (working concentration 0.1 μg/ml for WB, Sigma, T9026).

**N-glycosylation analysis**. Cellular lysates were incubated with glycoprotein denature buffer (New England Biolabs) for 10 min at 100 °C and then chilled on ice. The denatured glycoprotein was incubated with 500 U of EndoH (New England Biolabs) or PNGaseF (New England Biolabs) for 1 h at 37 °C, as per the manufacturer's instructions.

**Proteasome inhibition**. Ten-cm dishes of $Pkd2^{lrm4/lrm4}$ and $Pkd2^{+/+}$ MEFs were grown to confluency. Half were treated for 10 h with 5 μm/ml MG132 and half were untreated. Detached and adherent cells were collected, pooled and lysed in RIPA buffer. The blots were imaged and analysed using the Image Studio package (Li-COR Bioscience). Samples were normalised to Tubulin. Three biological and two technical replicates were performed of each genotype.

**WB analysis**. Cell lysates were loaded on 4–12% Tris-acetate sodium dodecyl sulfate–polyacrylamide precast gels (Invitrogen) and transferred to polyvinylidene difluoride membrane (Bio-Rad). Membranes were blocked with 5% milk block and incubated with primary antibodies overnight. Membranes were washed with TBS-Tween 20 buffer and probed with fluorescent secondary antibodies prior to detection via a LI-COR Odyssey imaging system. Uncropped images for all WBs can be found in Supplementary Figs. 3–6 and 9.

**In silico analysis**. Human and mouse PC2 sequences were analysed using NetNGlyc1.0 in silico glycosylation site prediction software[69].

**Microscopy**. For SIM, fluorescently labelled cells were imaged using a Zeiss Super Resolution LSM 710 ELYRA PS.1. Images were collected from two sets of Wt, $Pkd2^{lrm4/+}$ and $Pkd2^{lrm4/lrm4}$ MEF cells, and the results were confirmed in embryonic kidney cells grown from two pairs of $Pkd2^{lrm4/lrm4}$ kidneys and two pairs of Wt kidneys. For confocal imaging, cells were prepared as above and imaged using a Zeiss LSM 700 inverted confocal microscope (×63 NA 1.4 PLANAPO).

**Reporting summary**. Further information on research design is available in the Nature Research Reporting Summary linked to this article.

## Data availability

The authors declare that the main data supporting the findings of this study are available within the article and its Supplementary Information files. Extra data, i.e. additional image sets are available from the corresponding author upon request. A source data file is available, which includes numbers used for densitometry in Figs. 2 and 5.

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

## Acknowledgements

We thank Jackie Harrison, Lucie Visor and Sara Wells (Mary Lyon Centre, Harwell) for animal husbandry and Jeremy Sanderson, Helen Hilton, Caroline Barker and Adele

Austin (Medical Research Council, Harwell) for technical assistance and advice. This work was supported by awards from the UK Medical Research Council to D.P.N. (MC_U142670370) and M.M.K. (MR/L002876/1) and by the NIDDK sponsored Baltimore Polycystic Kidney Disease Research and Clinical Core Center, P30DK090868 (University of Maryland School of Medicine Division of Nephrology, 655W. Baltimore St., Baltimore, MD 21201).

## Author contributions

J.L.K., D.T.G. and D.P.N. initially conceived the project. R.V.W. implemented the work with supervision from D.P.N. and J.L.K. D.J.W. performed qRT-PCR analysis. V.S. performed elements of additional analysis in response to reviewer comments. D.W., M.M.K. and R.V.W. undertook the structural image resolution microscopy and image analysis. C.E. provided supervision and training in protein biochemistry. The manuscript was drafted by R.V.W. and D.P.N. All authors critically reviewed the text.

## Additional information

**Competing interests:** The authors declare no competing interests.

