## [Peer Review File · Nature Communications]

Reviewers' Comments:

Reviewer #1:

Remarks to the Author:

In this paper, Rebecca Walker and colleagues investigated the ciliary role of Polycystin-2 (PC2) by analysing mice harbouring a non-ciliary localising, yet channel-functional Pkd2 mutation (PC2-W414G variant, Pkd2 Irm4 mouse model). The study design follows recent data that questioned the relationship and causal link between PC2 and the primary cilium in ADPKD.

In this study, mutant mice developed embryonic polycystic kidney disease with accumulation of PC2 at the ciliary base. Pkd2Irm4/Irm4 embryos developed kidney cysts that proved indistinguishable from those completely lacking PC2. The authors could show that reduced protein levels did not cause PKD per se as similar reductions were seen in heterozygous knockout mice that do not form kidney cysts. Thus, altered subcellular localisation of mutant PC2 was the most likely explanation for the ADPKD phenotype observed in Pkd2Irm4/Irm4 mice.

The authors could further demonstrate that ciliary localisation of the major ADPKD protein PC1 depends on PC2 ciliary entry which supports the recently published concept that both Polycystins enter the cilium as a complex, i. e. joining together before entering the cilium. However, as Walker and colleagues do acknowledge the question still remains how the Polycystins traffic to the base of the cilium, independently or in complex with each other.

The authors tackle an important aspect of PKD and add significant new data to the field providing strong evidence that cilia-localised PC2 functions as an anti-cystogenic signal, i. e. that the localisation of PC2 to the cilium seems to be necessary to prevent ADPKD. As PC2 expression solely in the cilium was not analysed, additional anti-cystogenic activity at other sites of the cell cannot be excluded, however.

I only have the following minor comments and questions:

Page 5: The orpk mouse carries biallelic mutations in Ift88 and thus might not be a good model for PKD.

Page 7: Please state if the null allele of Pkd2 you examined and refer to has the same genetic background (C57BL/6J) as your mutant mice described.

Figure 2F: May I ask if the cell morphology of the mutant cells is altered?

Page 14: You mention "less intense PC2 ring in wildtype", please add further data and/or statistical evaluation.

Figure 4: Did you quantify Arl13b? The signal looks weaker in mutant cells.

Carsten Bergmann

Reviewer #2:

Remarks to the Author:

Walker and colleagues present a study in which a PC2 mutant that retains channel activity but is unable to enter the ciliary compartment is found to recapitulate the loss of function phenotype with respect to kidney cyst formation in vivo. They conclude from this that the critical function of PC2 related to preventing cyst formation requires cilia location of PC2.

While the authors are correct to raise plausible questions about the existing evidence confirming the essential role of cilia location for PC2 function, the likelihood of this not being the case based on the extensive existing literature is low. That said, a definitive demonstration that the cilia location in and of itself is essential to the function of PC2 in preventing PKD is valuable. The crux of the interpretation of the data in this manuscript rests on the premise that the PC2 point mutant described has normal channel activity. The authors cite a 2012 Science paper in which ER microsome bilayer studies were used to evaluate the channel activity of this mutant.

Unfortunately, since that time, there has been significant controversy in the evaluation of PC2 functional channel properties. The reported channel properties of the protein evaluated by direct patch clamping of cilia membranes is different from the channel properties reported by other

methods and the functional stoichiometry of the PC2 channel in the ER versus cilia membrane (homotetramer versus PC1-heterotrimer) further complicates interpretation. Indeed some of the structural studies have suggested that the TOP domain where the mutation resides may have some role in the channel properties of PC2 (a cation “antechamber”) as well as in PC1-PC2 complex formation.

Overall the trafficking studies in this manuscript are very nicely described and offer a significant contribution but the interpretation that this “proves” that cilia location is essential for the function of PC2 cannot be categorically stated. Therefore, the authors should describe unique trafficking properties of this mutant and suggest that this is a further indication, if the channel activity is indeed preserved, that the cilia location is an essential component of PC2 function. The manuscript is therefore an elegant description of the trafficking of a specific PC2 mutant and adds to the mass of evidence supporting the importance of cilia location for PC2 function in averting PKD but is not definitive in this regard and so should be presented in this context.

Major comments:

1. Although nothing to be done about it at present (we are not asking that this experiment be done), for the in vivo phenotyping (Fig.1) the authors who would have been better served to produce a Cre recombinase model in which the *Pkd2Irm4* allele was paired with a *Pkd2flox* allele so that they could conditionally delete the latter specifically in later stage kidneys and compare it to the *Pkd2flox/null* mouse kidney. This would have isolated the relative phenotypes of the missense and null mutants to the kidney only and would have eliminated any confounding features resulting from loss of function in other organs during early development—e.g., the embryonic lethality is not the result of the kidney phenotype. Such a model would be more instructive for subtle differences in kidney related severity.

2. The authors did an impressive amount of work in obtaining early stage embryonic kidneys and producing lysates to quantitate protein expression (Fig. 2). That said, this is a complicated experiment to interpret. The wild type versus the heterozygous kidney is relatively straightforward as both develop normally and have normal tissue architecture and there is a 50% reduction in protein levels as would be expected from hemizygoty. However the mutant homozygous kidneys are not normal in structure and therefore the 50% reduction in protein levels could represent relatively less cells expressing the protein, cells that have altered phenotypes that includes lower levels of PC2 expression, altered transcription or more likely mRNA stability, or altered quality control and processing of the post-translational product. On the face of it one cannot say that the reduction in protein level does not have a role in the pathogenesis of this mutant since reduced levels of a partially functioning protein may be sufficient to elicit a phenotype whereas reduced levels of fully functional protein (the heterozygous state) do not. Therefore the section title statement “reduced protein levels are not responsible for cyst development” is an overstatement. The most that could be said is that “reduced protein levels are not solely responsible for.....”

3. The Western blots in figure 2 have the added complication that almost all of the blots with the mutant protein seem to show PC2 as a doublet or at least as a different migration pattern from the wild type. This is true in panel A, panel E as well as the respective supplemental figures. This suggests added complexity to effects of this missense change and the authors should at least incorporate this observation into their interpretation of data.

4. As the authors clearly discuss, the Endo H studies offer limited additional insights although they are an important control for the study.

5. The SIM localization data (Figs. 3 and 4) for wild type and mutant PC2 as well as for PC1 in the context of mutant PC2 and the demonstration that both wild type or mutant PC2 are associated with Cep164 distal appendages are novel and interesting contributions.

Minor comments

1. It would be good to document Pkd2 mRNA levels by quantitative RT-PCR in the mutant and control embryonic kidneys to correlate with the protein expression data.
2. There are published experimental data on PC2 N-glycosylation that the authors should incorporate in addition to their computational predictions.
3. The discussion of the potential trafficking defect resulting from the TOP domain mutation should be expanded to include possibilities such as: a) the mutation affects the co-assembly of three PC2 molecules with a single PC1 molecule as suggested by recent structural studies and raise the possibility that these heterodimeric complexes are required to enter cilia; b) the TOP mutation results in subtle channel defects (as noted above) and normal channel function is required for cilia entry.
4. Although not the focus of this paper, some cell based studies regarding the stability of the PC2Irm4 mutation would have been interesting. For example does proteasome inhibition increase steady-state levels in primary cell cultures as indication that this protein is more prone to degradation as suggested in the discussion?

Reviewer #3:

Remarks to the Author:

The paper of Rebecca Walker et al., dissects the contribution of PC2 in PKD by using a non-ciliary localizing, yet channel-functional mutant version of PC2 (E442G). They show that the localization of PC2 to the cilium is in fact a critical factor in preventing PKD. The results are interesting and well presented but in its current form the paper lacks the mechanistic aspect.

P9: PC2 dosage? What does it mean ?

The paragraph title is too general while it only focuses on the abundance of PC2 (E442G).

Graph 2B: this is the Abundance of PDK2 protein normalized to tubulin.

The diagram depicting cleavage sites is wrong for EndoH. EndoH cleaves hybrid type glycan structures as well as oligomannose type structures. The action of EndoH requires that the mannose in alpha 1,6 is substituted by two other mannose residues.

P12, the sentence "PC2wt and PC2Irm4 protein were EndoH sensitive, consistent with them not passing through the Golgi" is not completely true as hybrid type glycan structures (structures acquired when glycoproteins move through the Golgi apparatus) can be cleaved by EndoH. The result mainly suggests that the trimming of mannose residues is impaired either due to lack of Golgi targeting of PC2 or dysfunction in Golgi mannosidases. This part has to be tempered.

The authors highly suggest that the point mutation impacts the folding of the glycoprotein and that PC2Irm4 is degraded via ERAD. The use of MG132 that block proteasomal degradation would definitely answer that point.

The immunofluorescence experiment shown in Fig 2 is too vague to conclude something. Several markers should be used as ER marker as well as plasma membrane marker in different conditions (with and without MG132).

P17. The lack of PC1 ciliary localization is very interesting. The authors should show the steady state level of PC1 in PC2 Irm4. This is possible that the complex PC1/PC2 occurs in the ER when these two proteins are synthesized. The instability of one can impact the stability of the other. It

would also be very interesting, as PC1 is also N-glycosylated, to look at EndoH/ PNGase sensitivity. It's possible that PC1 traffics through the Golgi at the opposite of PC2.

We thank the reviewers for their positive comments on the manuscript, which we found both insightful and constructive. Reviewer 1 states that “the authors tackle an important aspect of PKD and add significant new data to the field providing strong evidence that cilia-localised PC2 functions as an anti-cystogenic signal, i. e. that the localisation of PC2 to the cilium seems to be necessary to prevent ADPKD.” Reviewer 2 states that “Overall the trafficking studies in this manuscript are very nicely described and offer a significant contribution” and “The manuscript is therefore an elegant description of the trafficking of a specific PC2 mutant and adds to the mass of evidence supporting the importance of cilia location for PC2 function in averting PKD but is not definitive in this regard”. Reviewer 3 states that “They show that the localization of PC2 to the cilium is in fact a critical factor in preventing PKD. The results are interesting and well presented but in its current form the paper lacks the mechanistic aspect.”

Driven by these comments we have now made a number of changes to the manuscript. We particularly wish to stress the following additional data.

Further experiment 1 Analysis of *Pkd2* mRNA levels

We have analysed *Pkd2* expression by qRT-PCR, demonstrating that mRNA expression levels do not underlie reduced protein levels. Briefly, two biological replicates per genotype were analysed – further, a technical replicate was performed for each sample. While these studies demonstrated a clear reduction in mRNA levels in the *Pkd2*^{-/-} (null mutant), no reduction was evident in expression in *Pkd2*^{lrm4/lrm4} cells.

A supplementary figure (S10) shows the qPCR results. A schematic representation of the wildtype, indicating the position of the *lrm4* point mutation, and engineered null locus (derived by the Watnick group and previously demonstrated to be protein null). The three ensembl annotated *Pkd2* transcripts are also shown. The Exon 8-9 and 12-13 qRT-PCR assays can only identify the protein-coding *Pkd2*-201 transcript.

Further experiment 2 Analysis of PC1 protein levels.

We have analysed PC1 levels in *Pkd2*^{lrm4/lrm4}, *Pkd2*^{+/-} and *Pkd2*^{+/+} MEFs. We find that PC1 levels in *Pkd2*^{lrm4/lrm4} cells are very similar to those in *Pkd2*^{+/-} cells, consistent with PC1 levels in these cells simply reflecting the level of PC2 protein. Notably PC1 expression rises with loss of PC2, such that the highest expression levels are evident in *Pkd2*^{-/-} cells. This data has been added to Figure 4.

Further experiment 3 Analysis of proteasome inhibition on PC2^{lrm4} protein levels

To test whether PC2^{lrm4} is degraded by the proteasome we inhibited proteasome function to see whether PC2^{lrm4} levels could be restored to wildtype levels. A small increase in both wildtype and PC2^{lrm4} absolute protein levels was detected following treatment, but PC2^{lrm4} protein levels did not get close to WT levels, indicating that proteasomal degradation is not fully responsible for the differences in protein levels. In light of the qRT-PCR results indicating that mRNA levels are similar in Wt and *Pkd*^{lrm4/lrm4} cells, the reason for the reduced PC2^{lrm4} protein level remains elusive.

We feel it important to note that the H280 PC-2 antibody (Santa Cruz Biotechnology) used in this study was withdrawn (together with most or all other Santa Cruz Biotechnology polyclonal antibodies). This followed USDA action against Santa Cruz Biotechnology (see <https://www.nature.com/news/us-government-issues-historic-3-5-million-fine-over-animal-welfare-1.19958>). We ordered and stored extra reagent to complete this study, but finished our final aliquot in producing the additional analysis – this limited the number of experiments we were able to perform. We have tested the other commercially available PC2 antibodies and find them not to work in our hands. We are in the process of raising new PC2 antibodies, but any resulting reagent is not yet available.

Reviewer #1 (Remarks to the Author):

In this paper, Rebecca Walker and colleagues investigated the ciliary role of Polycystin-2 (PC2) by analysing mice harbouring a non-ciliary localising, yet channel-functional Pkd2 mutation (PC2-W414G variant, Pkd2 Irm4 mouse model). The study design follows recent data that questioned the relationship and causal link between PC2 and the primary cilium in ADPKD.

In this study, mutant mice developed embryonic polycystic kidney disease with accumulation of PC2 at the ciliary base. Pkd2Irm4/Irm4 embryos developed kidney cysts that proved indistinguishable from those completely lacking PC2. The authors could show that reduced protein levels did not cause PKD per se as similar reductions were seen in heterozygous knockout mice that do not form kidney cysts. Thus, altered subcellular localisation of mutant PC2 was the most likely explanation for the ADPKD phenotype observed in Pkd2Irm4/Irm4 mice.

The authors could further demonstrate that ciliary localisation of the major ADPKD protein PC1 depends on PC2 ciliary entry which supports the recently published concept that both Polycystins enter the cilium as a complex, i. e. joining together before entering the cilium. However, as Walker and colleagues do acknowledge the question still remains how the Polycystins traffic to the base of the cilium, independently or in complex with each other.

The authors tackle an important aspect of PKD and add significant new data to the field providing strong evidence that cilia-localised PC2 functions as an anti-cystogenic signal, i. e. that the localisation of PC2 to the cilium seems to be necessary to prevent ADPKD. As PC2 expression solely in the cilium was not analysed, additional anti-cystogenic activity at other sites of the cell cannot be excluded, however.

I only have the following minor comments and questions:

Page 5: The orpk mouse carries biallelic mutations in Ift88 and thus might not be a good model for PKD.

We have now altered the text to read: "The Oak Ridge Polycystic Kidney mouse was one of the earliest demonstrations of the importance of primary cilia in renal cyst development"

Page 7: Please state if the null allele of Pkd2 you examined and refer to has the same genetic background (C57BL/6J) as your mutant mice described.

All of this work was performed on mice congenic for C57BL/6J. We have now made this clear in our Materials and Methods: "*Pkd2*^{Irm4} mice were congenic on C57BL/6J. The *Pkd2* allele, *Pkd2*^{Irm1.2Tjw}, was derived from *Pkd2*^{Irm1.1Tjw} by Cre deletion and maintained congenic on C57BL/6J."

Figure 2F: May I ask if the cell morphology of the mutant cells is altered?

We understand why the images presented lead the reviewer to ask this question. In fact we observed no overall alteration in cell morphology, either in live cells or fixed cells. We have altered the images that we are showing in Figure 2F as well as adding Figure 2G to more closely reflect this.

Page 14: You mention "less intense PC2 ring in wildtype", please add further data and/or statistical evaluation.

We have further analysed images where the ring of PC2 is apparent at the base of the cilium. We have included several images in Fig S8 of MEF cilia from Wt and Irm4/Irm4 cells where the ring of PC2 is clearly evident at the base of the cilium. The intensity of the PC-2 channel-positive pixels within a 1µm dia circle at the ciliary base was measured. Although the mutant rings were on average more intense (13.08 vs 10.74 mean intensity value), this was not statistically significant, at least for the sample available. Therefore, we have altered our wording to state "an often less intense ring in Wt" can be noted.

Figure 4: Did you quantify Arl13b? The signal looks weaker in mutant cells.

While we agree with the reviewer that this is an interesting question, we did not quantify Arl13b believing it to fall beyond the scope of this investigation. We do not feel that we have sufficient data to answer this question. Links have been previously been established between Arl13b and PC2, including the recent Hwang et al paper (<https://www.ncbi.nlm.nih.gov/pubmed/30799239>), on which Dr Walker is an author.

--

Reviewer #2 (Remarks to the Author):

Walker and colleagues present a study in which a PC2 mutant that retains channel activity but is unable to enter the ciliary compartment is found to recapitulate the loss of function phenotype with respect to kidney cyst formation *in vivo*. They conclude from this that the critical function of PC2 related to preventing cyst formation requires cilia location of PC2.

While the authors are correct to raise plausible questions about the existing evidence confirming the essential role of cilia location for PC2 function, the likelihood of this not being the case based on the extensive existing literature is low. That said, a definitive demonstration that the cilia location in and of itself is essential to the function of PC2 in preventing PKD is valuable. The crux of the interpretation of the data in this manuscript rests on the premise that the PC2 point mutant described has normal channel activity. The authors cite a 2012 Science paper in which ER microsome bilayer studies were used to evaluate the channel activity of this mutant. Unfortunately, since that time, there has been significant controversy in the evaluation of PC2 functional channel properties. The reported channel properties of the protein evaluated by direct patch clamping of cilia membranes is different from the channel properties reported by other methods and the functional stoichiometry of the PC2 channel in the ER versus cilia membrane (homotetramer versus PC1-heterotrimer) further complicates interpretation. Indeed some of the structural studies have suggested that the TOP domain where the mutation resides may have some role in the channel properties of PC2 (a cation "antechamber") as well as in PC1-PC2 complex formation.

Overall the trafficking studies in this manuscript are very nicely described and offer a significant contribution but the interpretation that this "proves" that cilia location is essential for the function of PC2 cannot be categorically stated. Therefore, the authors should describe unique trafficking properties of this mutant and suggest that this is a further indication, if the channel activity is indeed preserved, that the cilia location is an essential component of PC2 function. The manuscript is therefore an elegant description of the trafficking of a specific PC2 mutant and adds to the mass of evidence supporting the importance of cilia location for PC2 function in averting PKD but is not definitive in this regard and so should be presented in this context.

Major comments:

1. Although nothing to be done about it at present (we are not asking that this experiment be done), for the *in vivo* phenotyping (Fig.1) the authors who would have been better served to produce a Cre recombinase model in which the Pkd2^{lrm4} allele was paired with a Pkd2^{flox} allele so that they could conditionally delete the latter specifically in later stage kidneys and compare it to the Pkd2^{flox}/null mouse kidney. This would have isolated the relative phenotypes of the missense and null mutants to the kidney only and would have eliminated any confounding features resulting from loss of function in other organs during early development—e.g., the embryonic lethality is not the result of the kidney phenotype. Such a model would be more instructive for subtle differences in kidney related severity.

We thank the reviewer for these insightful suggestions.

2. The authors did an impressive amount of work in obtaining early stage embryonic kidneys and producing lysates to quantitate protein expression (Fig. 2). That said, this is a complicated experiment to interpret. The wild type versus the heterozygous kidney is relatively straightforward as both develop normally and have normal tissue architecture and there is a 50% reduction in protein levels as would be expected from hemizygoty. However the mutant homozygous kidneys are not normal in structure and therefore the 50% reduction in protein levels could represent relatively less cells expressing the protein, cells that have altered phenotypes that includes lower levels of PC2 expression, altered

transcription or more likely mRNA stability, or altered quality control and processing of the post-translational product. On the face of it one cannot say that the reduction in protein level does not have a role in the pathogenesis of this mutant since reduced levels of a partially functioning protein may be sufficient to elicit a phenotype whereas reduced levels of fully functional protein (the heterozygous state) do not. Therefore the section title statement “reduced protein levels are not responsible for cyst development” is an overstatement. The most that could be said is that “reduced protein levels are not solely responsible for.....”

We completely agree with the reviewer’s concerns about the cell types and numbers present in cystic versus non-cystic kidneys. As part of the work-up for this portion of the project we collected and analysed wildtype and *Irm4/Irm4* kidneys both at E14.5 and E15.5 – consistent with published data on PKD models we were unable to differentiate wildtype from mutant samples by histology at E14.5 (**we have added E14.5 kidney sections as Fig S2**). This led us to choose to collect and analyse E14.5 kidneys for these experiments. In light of the reviewer’s concerns the text now reads:

“We collected protein from E14.5 *Pkd2^{Irm4/Irm4}* kidneys; at this stage mutants showed normal renal architecture, no cysts and in our analysis proved indistinguishable from wildtype (Fig. S2). Collecting protein at this stage therefore avoids any confounding effects of comparing cystic to non-cystic kidneys.”

We have altered the section title to now read: “**Reduced PC2 abundance alone cannot explain cyst development in *Pkd2^{Irm4/Irm4}* embryonic kidneys**”

We have further added data addressing the mRNA levels and protein stability – please see **further experiment 1** at the beginning of our response.

On page 9 we now state: “

To understand this reduction in PC2 levels, we examined *Pkd2* expression by quantitative reverse transcription (qRT)-PCR. Using three sets of *Pkd2*-specific primers we detected no significant difference between wildtype and *Pkd2^{Irm4/Irm4}* MEF samples (Fig. S10), revealing that altered mRNA levels do not underlie the reduction in PC2 abundance. Next, we assessed whether PC2^{Irm4} was being preferentially degraded by the proteasome. We incubated MEF cells with the proteasome inhibitor MG132. In wildtype cells this led to an 11% increase in PC2 levels relative to untreated cells, whereas the *Pkd2^{Irm4/Irm4}* MEFs a 45% increase was seen over uninhibited PC2^{Irm4} levels. However, the absolute level of protein increase was similar between the two samples (Fig S11). Notably, proteasome inhibition was not sufficient to restore the PC2^{Irm4} abundance to wildtype levels, suggesting that reduced levels of PC2^{Irm4} cannot simply be explained by enhanced degradation via the proteasome. We cannot rule out increased degradation of PC2^{Irm4} via another pathway.

”

And we have discussed other possible mechanisms: “It is formally possible, for example, that PC2^{Irm4} protein exhibits a reduced, hypomorphic, function in addition to reduced protein levels. In such a situation the level of PC2 function might drop beneath that required to prevent cyst formation. To investigate this would require the development of a measurable assay of PC2’s anti-cystic function, or a correlate of this function; this in turn would require a deeper understanding of the mechanism by which PC2 prevents cyst formation.”

3. The Western blots in figure 2 have the added complication that almost all of the blots with the mutant protein seem to show PC2 as a doublet or at least as a different migration pattern from the wild type. This is true in panel A, panel E as well as the respective supplemental figures. This suggests added complexity to effects of this missense change and the authors should at least incorporate this observation into their interpretation of data.

We believe that the banding pattern in *Pkd2^{Irm4/Irm4}* samples does indeed reflect the Wt pattern and suggest that the band may run as a doublet in the Wt sample but is not resolved due to the higher expression levels. It is possible that in *Irm4*, both the abundance and ratio of these bands is changed. This could reflect the altered cellular localisation that we see of PC2^{Irm4}.

We have discussed this on page 23 “The lowered expression level in *Pkd2*^{lrm4/lrm4} samples reveals an altered abundance or ratio of bands compared to the Wt banding pattern. This may indicate that PC2^{lrm4} has altered cellular trafficking.”

4. As the authors clearly discuss, the Endo H studies offer limited additional insights although they are an important control for the study.

5. The SIM localization data (Figs. 3 and 4) for wild type and mutant PC2 as well as for PC1 in the context of mutant PC2 and the demonstration that both wild type or mutant PC2 are associated with Cep164 distal appendages are novel and interesting contributions.

Minor comments

1. It would be good to document *Pkd2* mRNA levels by quantitative RT-PCR in the mutant and control embryonic kidneys to correlate with the protein expression data.

We have further analysed mRNA levels by qPCR in light of these comments – see **further experiment 1** at the beginning of our response.

2. There are published experimental data on PC2 N-glycosylation that the authors should incorporate in addition to their computational predictions.

We thank the reviewer for pointing this out this embarrassing omission. We have altered the text to now read: “Using *in silico* analysis we were able to identify nine putative N-glycosylation sites in PC2, five falling within the first extracellular loop (S1-S2 loop; aa243-468) (Fig. 2.D), in which the *Pkd2*^{lrm4} point mutation (E442G) lies; these five sites were previously identified experimentally (44)”

44: Hofherr A, Wagner C, Fedeles S, Somlo S, Köttgen M. N-glycosylation determines the abundance of the transient receptor potential channel TRPP2. The Journal of biological chemistry. 2014;289(21):14854-67.

3. The discussion of the potential trafficking defect resulting from the TOP domain mutation should be expanded to include possibilities such as: a) the mutation affects the co-assembly of three PC2 molecules with a single PC1 molecule as suggested by recent structural studies and raise the possibility that these heterodimeric complexes are required to enter cilia; b) the TOP mutation results in subtle channel defects (as noted above) and normal channel function is required for cilia entry.

We have now altered the discussion of the TOP domain on page 23 to include these arguments. “ In combination with our findings, this suggests that the S1-S2 loop region is required for normal transport of PC2 into the cilium. It is interesting to speculate that this might be through interference with PC1-PC2 heterotetramer formation and that mutations in the TOP domain directly influence these interactions. Investigation of the TOP domain structure reveals that the TOP domain assumes a different conformation in membranes with different lipid compositions (60). It is possible therefore that certain TOP domain mutations may alter the conformation sufficiently to prevent the channel from forming; such a fundamental block to complex formation could conceivably prevent PC1-PC2 complex trafficking. Conversely, mutations in the TOP domain may affect function of the channel complex, resulting in subtle alterations in channel activity that are not evident in the cell free system in which PC2^{lrm4} was previously tested. These two hypotheses could be tested if an assay existed that could distinguish homomeric, heteromeric and single subunit forms of the protein. Disrupted channel formation or function may lead to PC2 protein being sent for degradation, which would be consistent with the lower level of PC2 protein that we see in the mutant cells.”

4. Although not the focus of this paper, some cell based studies regarding the stability of the PC2^{lrm4} mutation would have been interesting. For example does proteasome inhibition increase steady-state levels in primary cell cultures as indication that this protein is more prone to degradation as suggested in the discussion?

We have now analysed the effect of blocking proteasome activity with MG132 – see **further experiment 3** at the beginning of our response.

We found MG132 treatment to be insufficient to restore PC2^{lrm4} protein abundance to Wt levels, leading us to conclude that the reduced abundance of PC2 in *Pkd2^{lrm4/lrm4}* cells is not solely due to protein degradation via the proteasome. We have therefore altered our discussion accordingly:

“In wildtype cells this led to an 11% increase in PC2 levels relative to untreated cells, whereas the *Pkd2^{lrm4/lrm4}* MEFs a 45% increase was seen over uninhibited PC2^{lrm4} levels. However, the absolute level of protein increase was similar between the two samples (Fig S11). Notably, proteasome inhibition was not sufficient to restore the PC2^{lrm4} abundance to wildtype levels, suggesting that reduced levels of PC2^{lrm4} cannot simply be explained by enhanced degradation via the proteasome. We cannot rule out increased degradation of PC2^{lrm4} via another pathway..”

--

Reviewer #3 (Remarks to the Author):

The paper of Rebecca Walker et al., dissects the contribution of PC2 in PKD by using a non-ciliary localizing, yet channel-functional mutant version of PC2 (E442G). They show that the localization of PC2 to the cilium is in fact a critical factor in preventing PKD. The results are interesting and well presented but in its current form the paper lacks the mechanistic aspect.

P9: PC2 dosage? What does it mean ?

The paragraph title is too general while it only focuses on the abundance of PC2 (E442G).

Graph 2B: this is the Abundance of PDK2 protein normalized to tubulin.

The diagram depicting cleavage sites is wrong for EndoH. EndoH cleaves hybrid type glycan structures as well as oligomannose type structures. The action of EndoH requires that the mannose in alpha 1,6 is substituted by two other mannose residues.

P12, the sentence “PC2wt and PC2lrm4 protein were EndoH sensitive, consistent with them not passing through the Golgi” is not completely true as hybrid type glycan structures (structures acquired when glycoproteins move through the Golgi apparatus) can be cleaved by EndoH. The result mainly suggests that the trimming of mannose residues is impaired either due to lack of Golgi targeting of PC2 or dysfunction in Golgi mannosidases. This part has to be tempered.

The authors highly suggest that the point mutation impacts the folding of the glycoprotein and that PC2lrm4 is degraded via ERAD. The use of MG132 that block proteasomal degradation would definitely answer that point.

At the reviewer’s suggestion we have investigated proteasomal degradation in the *lrm4* cells. Please see **further experiment 3** at the beginning of our response and our response to reviewer 2 point 4 above.

The immunofluorescence experiment shown in Fig 2 is too vague to conclude something. Several markers should be used as ER marker as well as plasma membrane marker in different conditions (with and without MG132).

We agree with the reviewer that definitive conclusions from the staining that we show are challenging to make. This data is however, provided in support of the glycosylation analysis, which shows none of the differences reported for certain mis-trafficked PC2 mutant forms (aberrant cell surface trafficking, ER retention). No such mis-localisation is evident in the panels. We have added PDI staining to mark the ER (**Fig 2G**). As explained at the beginning of this response, the Santa Cruz Biotechnology Ab used for this work is no longer available and our stock was finished during the analysis. We have altered our wording to now state that: “These data suggest that there is no detectable global mis-targeting of the mutant protein, such as the aberrant cell surface trafficking which is seen with C-terminal deletion mutant constructs (47), a result supported by immunofluorescent imaging of whole cells (Fig. 2.F,G).”

and added: “We saw no overall alterations of cell morphology nor indications of PC2 accumulation in the ER (Fig. 2. G) of *Pkd2^{lrm4/lrm4}* MEFs.”

P17. The lack of PC1 ciliary localization is very interesting. The authors should show the steady state level of PC1 in PC2 *lrm4*. This is possible that the complex PC1/PC2 occurs in the ER when these two proteins are synthesized. The instability of one can impact the stability of the other. It would also be very interesting, as PC1 is also N-glycosylated, to look at EndoH/ PNGase sensitivity. It's possible that PC1 traffics through the Golgi at the opposite of PC2.

We thank the reviewer for their insight in suggesting this line of enquiry. **Further experiment 2** at the start of this response describes our analysis of PC1 and PC2 levels in cells, revealing that while there is a rise in PC1 levels on the loss of PC2 (*Pkd2*^{-/-}), this is not mirrored in *lrm4* mutation, which leads to a result similar to *Pkd2*^{+/-}. Therefore PC1 levels are unaffected by *lrm4*, beyond the effect due to protein level changes. We agree that further analysis of the glycosylation would be intriguing, but that falls beyond the scope of this study and we feel that we need to correlate any such alterations with equivalent analysis of PC2 in the same samples. As such this will have to wait until we have a new PC2 antibody.

Reviewers' Comments:

Reviewer #1:

Remarks to the Author:

no further comments

Reviewer #2:

Remarks to the Author:

Overall, the authors have done a good job of addressing comments from the previous review. The manuscript presents two salient points:

First, that the PC2Irm4 mutant has a selective defect in entering cilia. Such mutants have been described before, but the important addition from this work is demonstration that this mutant accumulates at the ciliary base in the region of the mother centriole. This is the first demonstration of a steady state intracellular location for PC2 other than the ER.

Second, this mutant is a complete loss of function. The conclusion that this is due to the ciliary trafficking defect rather than some other hypomorphic effect is inferred, not proven, but proof in this case is challenging without a better quantitative cilia-independent functional readout for PC2. The authors now point this out.

Overall, this work will enhance the field and should interest the readership with the data for the above points.

A minor comment:

In the discussion, the statement that "In this study we demonstrate that PC2Irm4 similarly fails to localise to the ciliary axoneme in the developing kidney and in fibroblasts" needs to be modified. Unless I missed it, nowhere does this manuscript address the cilia location of t PC2Irm4 in the kidney.

Reviewer #3:

None

Response to reviewer comments:

REVIEWERS' COMMENTS:

We should like to thank all three reviewers. Their universally constructive comments and smart questions have, in our opinion, significantly improved this manuscript.

Reviewer #1 (Remarks to the Author):

no further comments

--

Reviewer #2 (Remarks to the Author):

Overall, the authors have done a good job of addressing comments from the previous review. The manuscript presents two salient points:

First, that the PC2^{lrm4} mutant has a selective defect in entering cilia. Such mutants have been described before, but the important addition from this work is demonstration that this mutant accumulates at the ciliary base in the region of the mother centriole. This is the first demonstration of a steady state intracellular location for PC2 other than the ER.

Second, this mutant is a complete loss of function. The conclusion that this is due to the ciliary trafficking defect rather than some other hypomorphic effect is inferred, not proven, but proof in this case is challenging without a better quantitative cilia-independent functional readout for PC2. The authors now point this out.

Overall, this work will enhance the field and should interest the readership with the data for the above points.

A minor comment:

In the discussion, the statement that "In this study we demonstrate that PC2^{lrm4} similarly fails to localise to the ciliary axoneme in the developing kidney and in fibroblasts" needs to be modified. Unless I missed it, nowhere does this manuscript address the cilia location of t PC2^{lrm4} in the kidney.

We agree that the comment in the discussion could have been misleading as we have indeed not shown kidney sections in our PC2 ciliary localisation analysis. Instead, we imaged outgrowths of embryonic kidneys to show that PC2^{lrm4} is unable to localise along the ciliary axoneme. We have accordingly modified the wording in the discussion to be more precise:

"In this study we demonstrate that PC2^{lrm4} similarly fails to localise to the ciliary axoneme in embryonic kidney outgrowths and in fibroblasts."

<ALG: please note Referee #3 provided confidential comments to us:

"The authors have satisfactorily answered my comments. In its current form, the paper is now suitable for publication." >

We would particularly like to thank Reviewer #3 for pointing us towards a number of additional informative experiments.

We would also like to thank the editor for their comments and feedback regarding the manuscript.